# Do Animal Welfare Education Campaigns Really Work? An Evaluation of the RSPCA’s #DogKind Campaign in Raising Awareness of Separation-Related Behaviours in UK Dog Owners

**DOI:** 10.3390/ani14030484

**Published:** 2024-02-01

**Authors:** Izzie Philpotts, Emily J. Blackwell, Justin Dillon, Nicola J. Rooney

**Affiliations:** 1Animal Behaviour and Welfare Group, Bristol Veterinary School, University of Bristol, Bristol BS40 5DU, UK; emily.blackwell@bristol.ac.uk (E.J.B.); nicola.rooney@bristol.ac.uk (N.J.R.); 2School of Sport, Rehabilitation and Exercise Sciences, University of Essex, Wivenhoe Park, Colchester CO4 3SQ, UK; 3IOE, UCL’s Faculty of Education & Society, University College London, London WC1H 0AL, UK; justin.dillon@ucl.ac.uk

**Keywords:** animal welfare, behaviours, campaigns, dog, education, knowledge, separation-related behaviours, SRB, understanding

## Abstract

**Simple Summary:**

Animal charities want to teach people about what animals need and often run campaigns to achieve this. These campaigns usually focus on different aspects of animal welfare. In March 2019, the Royal Society for the Prevention of Cruelty to Animals (RSPCA) launched a campaign called #DogKind. It aimed to increase awareness of dog owners to identify separation-related behaviours in dogs and encourage owners to seek help from reliable sources. To evaluate the campaign’s effectiveness, we asked people a series of questions before and after the campaign. We also tested whether adding a video to the website’s information made any difference to the campaign’s effectiveness. This study found that the campaign successfully reached its target audience of 25–34-year-olds but did not help dog owners become more aware of separation-related behaviours or increase the number of owners intending to seek help from reliable sources. Additionally, showing a video as part of the campaign did not improve its effectiveness. This study shows that this campaign had limited success in achieving its targets and highlights the importance of thorough evaluations of campaigns as educational interventions.

**Abstract:**

One of the main aims of companion animal welfare charities is to educate the public about the needs of animals. This is frequently performed through campaigns focusing on specific aspects of welfare. The Royal Society for the Prevention of Cruelty to Animals (RSPCA), Britain’s biggest animal welfare charity, launched the nationwide #DogKind campaign in March 2019. Targeted mainly at 25–34-year-olds, the campaign aimed to increase awareness of separation-related behaviour (SRB) among dog owners and encourage them to seek help for SRB from reliable sources. This research involved a quasi-experimental, non-equivalent control group design evaluating the campaign’s effectiveness. It was conducted through a series of online surveys at three different time points: before the launch of the campaign (n = 2002), six months after (n = 2423), and, again, two months later (n = 269), during which we asked the same questions regarding knowledge of SRB. An experimental trial of 269 participants tested whether accessing a video alongside the campaign web pages increased the effectiveness of the campaign objectives. Overall, the campaign appeared to be effective in reaching its target audience but not at raising awareness of SRB or increasing the number of owners intending to seek help. The inclusion of a video in the campaign made no difference to its effectiveness. This study shows that this campaign had limited success in achieving its targets and highlights the importance of thorough evaluations of education interventions that aim to improve the welfare of companion animals.

## 1. Introduction

Education is central to the work of companion animal welfare charities in the United Kingdom (UK) [1,2,3]. They aim to educate the public on aspects of an animal’s health, needs, and behaviours to improve their welfare. Education through targeted marketing campaigns is one key method by which charities aim to heighten public awareness of specific issues. Public campaigns use an organised set of communication strategies, typically through media and messaging, to deliver information to a targeted population over a defined period to achieve a specific set of outcomes [4]. A substantial amount of a charity’s money and other resources are spent on the creation and running of these campaigns (e.g., the RSPCA spent GBP 5.7 M in 2022 on campaigns and education [5]), yet there remains very limited evidence in the public domain of either their outcomes or effectiveness in improving the welfare of companion animals.

### 1.1. What Do We Know about Campaigns?

Campaigns utilise multiple different theories, frameworks, and strategies that have evolved over the last 50 years or more; however, no one specific approach has yet been developed to explain and predict their effectiveness [4]. Many campaigns, including health, prosocial, and environmental, share significant similarities and approaches to that of commercial advertising [6] and can be an effective source of messaging. For example, public health campaigns have reduced drunk driving and smoking and increased healthy eating and blood donation [7,8]. While many campaigns involve long-term awareness raising, others have shown positive impacts, albeit modest, with interventions lasting just two months [9]. In a systematic review of the use of mass media communication of public health messages in six health topic areas, Stead et al. [10] reported that longer and more intensive campaigns were likely to be more effective.

Many campaigns aim to invoke a cognitive or emotional response at an individual level, impacting decision-making processes and ultimately facilitating changes in people’s behaviour [11]. Changing behaviour can also occur through less direct routes, for example, by triggering discussions within social and professional networks [12]. Campaigns that reach large audiences can change the norms and expectations within groups through the phenomenon of social contagion, so even individuals not directly exposed to the campaign may be influenced by their peers’ changes in attitudes and behaviours [7]. Ultimately, some campaigns can lead to changes in public policy at a societal level [7,13]. As an example, the charity Mind launched a public health campaign that led to local councils having to report how much of their public health budgets are spent on mental health [14].

Campaign development is a complex process involving multiple factors, including the target audience, the marketing approach and method, as well as the communication channels and dissemination strategies utilised. All these factors can influence the success of a campaign, whether that be judged by the number of people aware of the issue or, more importantly, long-term changes in behaviours [15]. The evaluation process of any campaign should also be considered, with clear performance indicators and appropriate evaluation strategy determined from the outset [4,16]. Interestingly, Rice and Atkin [4] concluded that most contemporary public communication campaigns have only a modest impact. A more recent series of reviews by Stead et al. [10] found very mixed and weak or limited evidence but did conclude that impact varied with the health topic addressed but rarely exceeded moderate success. The review also highlighted the challenges of synthesising campaign evaluations due to variations in approaches to measuring success and evaluation.

Campaigns targeted at improving the welfare of companion animals have taken place over many years, but whilst evaluations of their effectiveness may have been taking place internally, very few in the UK appear to have been published. Those that have been published include education interventions targeting school children [17,18,19], dog bite prevention interventions [17], cat neutering campaigns [20], and an evaluation of the efficacy of written advice in reducing separation-related behaviour (SRB) [21]. All these studies showed significant improvements in outcome measures following the education intervention compared to controls.

In addition, animal welfare charities routinely use powerful visual messaging in television advertisements and campaign videos. Whilst videos appear to provide both emotive and impactful messaging and have been used with some success in health campaigns [22], a formal evaluation of the effectiveness of videos in companion animal welfare campaigns has yet to be published. A recent scoping review of studies of the use of digital video interventions in mental health promotion found that results were generally positive but did note that involving end-users in co-creation of the materials was beneficial [23]. The reviewers also commented that there is a need for active involvement of end-users in co-creation and to attend to the production quality so that the digital video intervention is as relevant, informed, and effective as possible. Curran et al. [24] examined the impact of the “What’s up With Everyone?” campaign, which used video animations, on the mental health awareness of young people. They reported that knowledge, attitudes, confidence, and willingness to seek support improved post-test and that there were also significant reductions in the stigma towards depression. Video as a learning medium is widely used in educational settings and provides an effective information delivery tool, with students finding it engaging and often helpful to visualise more challenging concepts [25].

### 1.2. What Are Separation-Related Behaviours?

SRBs are unwanted canine behaviours that only occur in the absence or perceived absence of their owners [26]. The most frequently reported behaviours associated with separation from the owner are vocalisation, destructive behaviour, and inappropriate elimination [27,28]. Less frequently reported behaviours include self-mutilation, repetitive behaviour, excessive salivation, restlessness, vomiting and diarrhoea, and aggression toward the owner at the time of departure [29,30]. The exact number of dogs suffering from SRBs in the UK is somewhat difficult to determine as the behaviour is, by its nature, displayed only when the owner is not present, so figures based on owner reports are likely an underestimation of the true extent of the problem [21]. A longitudinal study of puppies conducted more than 20 years ago reported that over 50% of dogs had displayed some signs of SRBs by 18 months of age, although many of these were temporary and resolved spontaneously [31]. A point sample survey of dog walkers in Southern England showed that 13% of dogs in the general population exhibited apparent signs of SRBs at a given time and another 11% had in the past [32]. More recently the PDSA Paw Report [33] showed that 18% of owners who obtained their pet after March 2020 reported that their dog showed signs of distress when left. For owners who had their pets before March 2020, 5% reported new signs of distress when leaving their dog [33]. These findings are also reflected in the repeat measures survey conducted during lockdown and once restrictions had eased, which showed that 9.9% of dogs (n = 1807) developed SRBs once restrictions had eased and that the dogs whose leaving hours decreased most during lockdown were more at risk of developing SRBs [34]. This rise is perhaps to be expected, given that habituation to periods of social isolation during early life is important in preventing the development of SRB, which may have been less likely during the pandemic.

SRB is often a factor contributing to owners relinquishing their dogs. In the UK, Diesel et al. [35] showed that undesired behaviours were the most common reason given for relinquishment. These findings have been reflected in other studies both within the UK [36] and elsewhere [36,37,38,39]. In addition to the problems faced by owners in managing SRBs, the dog’s welfare is of significant concern. Dogs that display SRBs are often in a compromised emotional state with anxiety playing a significant factor [40]. Research has also suggested that other negative affective states such as frustration, panic, fear, and boredom may also be associated with different types of SRBs [41]. With an increase in dog ownership and the associated changes brought by the recurrent lockdowns in the UK over the last two years, it appears feasible to suggest that SRBs remain, and may have recently become, even more of a widespread problem to both dogs and owners than previously considered.

## 2. Methods

The research used a quasi-experimental, non-equivalent control group design to evaluate the effectiveness of a nationwide education campaign conducted by an animal welfare charity targeting current dog owners living in the UK.

### 2.1. What Is the #DogKind Campaign?

In 2018 the Royal Society for Prevention of Cruelty to Animals (RSPCA) published “Being #DogKind: How in tune are we with the needs of our canine companions?” [42]. The report described the results of a survey commissioned by the RSPCA and undertaken by a market research company that sampled 3049 dog owners between 11th and 17th July 2017. Whilst other studies such as Rioja-Lang et al. [43], who took a modified Delphi approach, have found poor owner knowledge to be a major welfare issue, the #DogKind report concluded that, generally, most owners had a good understanding of their dog’s needs but that this did not necessarily result in ownership behaviours that were always in the best interests of their dogs. These findings ultimately led to the development of the #DogKind campaign. Before the launch of the campaign, the RSPCA commissioned a series of focus groups to provide greater insight into key areas that needed addressing by the campaign and to identify a target demographic group. The key objectives of the campaign were encouraging positive reward-based training throughout a dog’s life; increasing awareness around which emotions dogs can and cannot feel, developing resources and signposting owners to ensure they access the best advice, and addressing some of the issues that owners face when living with a dog. Separation-related behaviour was chosen as the focus for the first year of the campaign as this has consistently been seen to be one of the most prevalent, yet hidden, welfare issues affecting dogs, e.g., Ref. [44]. The owner demographic identified by the focus groups to be targeted by the campaign was 25–34-year-olds, one-third of whom reported that their dog had shown signs of SRB [42]. Focus group discussions also determined that video was the preferred source of information delivery for that target age group.

The RSPCA launched the nationwide #DogKind campaign in March 2019 (https://www.rspca.org.uk/adviceandwelfare/pets/dogs/kind (accessed on 29 January 2024)), a public-facing campaign that included social media promotion, RSPCA presence at large public events, web-based content, advertising materials such as leaflets and posters, and collaborations with other agencies. These promotions and events were all planned for within the first few months of the campaign launch.

### 2.2. What Did This Study Evaluate?

To establish the effectiveness of the campaign, we identified the following indicators guided by Fishbein and Ajzen’s Theory of Reasoned Action [45]: firstly, the extent to which the campaign reached the target audience of 25–34-year-olds; secondly, whether it raised the awareness of SRB in dog owners; and finally, whether the campaign increased the proportion of dog owners intending to seek help for SRB from reliable sources. In addition, we added a further experimental element to test whether accessing a video on SRB alongside the web pages increased the effectiveness of the campaign objectives compared to just written information. It was hypothesised that the video may increase the campaign effectiveness. To enable this comparison, the campaign webpages were set to pseudo-randomly direct traffic for a specified period to either written information on SRB or that same written information plus a one-minute video showing dogs displaying SRB (https://m.facebook.com/watch/?v=2140070196098559&paipv=0&eav=AfYDviEAJXu96GNpN9pC20QMm7hZft8FFBbBRciDnJN4ECT4ICF6qWtCn8u7eUA7vD4&_rdr (accessed on 29 January 2024)).

### 2.3. Population and Procedures

The research was conducted through a series of online surveys of UK dog owners at three different time points: immediately prior to the launch of the campaign (“Pre-survey”), six months after the launch of the campaign (“Six-month survey”), and again two months later, i.e., eight months after the launch of the campaign (“Eight-month survey”). All events promoting the campaign were planned for the first few months after the campaign launch, therefore six months was selected as an appropriate timeline to establish the effectiveness of the campaign.

From a population of incentivised survey panellists (who complete online surveys in exchange for points that can be converted to financial credit) known to the market research company, UK-based dog owners were identified and invited to complete the pre-survey in March 2019. The RSCPA commissioned a market research agency to carry out the first two surveys (pre- and six-month), requesting a sample size of 2000 respondents at each time point. Stratified sampling was employed to obtain representation from the subpopulations of gender, age groups, and regions based on a nationally representative sample. To gain a baseline measure of knowledge, the pre-survey was conducted by the market research company from 1 to 4 March 2019, just before the launch of #DogKind and reached 2002 owners.

Six months after the launch of the campaign, all 2002 pre-survey respondents were contacted by the market research company and asked to complete a second online survey (six-month survey) between 6 and 10 September 2019. The survey contained the same questions as in the pre-survey, with the addition of 14 questions asking about awareness of and engagement with the campaign. Overall, 988 people (49.4%) responded (Subpopulation A) whilst 1014 did not (Subpopulation B; Table 1). An additional sample of 1013 new respondents was then obtained from another equivalent population of online survey panellists and was invited to complete the survey between 13 and 17 September 2019. The same recruitment methods and stratification sampling approach as for the baseline were taken. Overall, 584 complete responses were received (Subpopulation C), but it later transpired that the other 429 respondents (Subpopulation D) had not consented to share personally identifiable data with third parties so only the data that had been consented to be shared from this population was made available to the researchers. To reach the specified target of 2000 responses, a final additional sample of 421 (Subpopulation E) was recruited and surveyed five to seven days later. However, for this sample, due to a system error, basic owner demographic information and responses to awareness of #DogKind were collected, but no dog demographic information nor current knowledge of SRB and ownership practices thereby limiting the analysis for which the sample could be used.

Eight months after the launch of #DogKind, an additional study to explore any differences in the effectiveness of a campaign with a video, compared to just the written information alone in the webpages, was conducted. To enable this comparison, the campaign webpages were set to pseudo-randomly direct traffic for a specified period to either written information on SRB or that same written information plus a one-minute video showing dogs displaying SRB. A total of 634 respondents from subpopulations A and C, who completed the six-month survey and reported not being aware of #DogKind, and who consented to contact, were emailed by the authors. This sample was directed to the campaign web pages via an email invitation and was subsequently asked to complete the same online questions as those who were aware of the campaign during the six-month survey. This phase took place between 29 October and 13 November 2019. In total, 296 responses (46.7%) were received and 269 (42.4%) could be linked by respondents’ email to previously collected data for analysis. Of these respondents, 190 were from subpopulation A and 79 from subpopulation C. Overall, 39.7% of respondents reported seeing the video, 47.6% did not, and 12.6% were not sure (n = 269).

### 2.4. Questionnaires Development and Content

All questions were developed by the authors and the RSPCA’s Campaign Manager. The questionnaires were piloted with a small convenience group of dog owners, and minor wording amendments were made based on feedback. The pre-survey baseline questionnaire included 34 questions covering five areas and took approximately 10 min to complete: respondent demographics; dog details; current ownership practices relevant to SRB; knowledge of SRB; and attitudes towards seeking help for SRB and problem behaviours. The same 34 questions were asked in the six-month survey, with the addition of 14 questions covering: awareness of #DogKind; engagement with #DogKind; history of visiting the #DogKind webpages; knowledge of SRB; ownership practices; as well as the intention to seek help for SRB. The six-month survey took approximately 15 min to complete. Respondents were also asked for consent to being contacted about the final element of the study carried out by University of Bristol researchers during the eight-month survey. During the eight-month survey, participants were asked for their email addresses to allow researchers to link their responses to previous demographics and behaviours collected during the six-month survey. In addition, they were asked the same questions as the six-month survey regarding: visiting the #DogKind webpage, whether they watched a video or not, changes to knowledge of SRB and ownership practices, and intention to change behaviours or seek help. All questions were mandatory and closed ended (see Table 2), but where appropriate, respondents were given the option of “other” and were then asked to provide details.

### 2.5. Ethical Approval

Ethical approval for this research was obtained from the University of Bristol Faculty of Health Science Research Ethics Committee on 30 January 2019. Reference number 81203.

### 2.6. Data Handling

Data received from the market research agency were screened and cleaned. Repeat respondents were identified and pre-survey, and six-month survey answers were linked by participant identification number. Responses from the eight-month survey were linked by email addresses to previous data. Anonymised data were entered into SPSS (version 25) for analysis.

### 2.7. Managing the Subpopulations

Having identified five distinct subpopulations, three of which (C, D, and E) resulted from a lack of standardisation in the sampling approach, we examined those subpopulations for statistically significant and biologically meaningful differences. Of the four subpopulations at the six-month survey (A, C, D, and E), statistically significant differences in all six variables tested were found between subpopulations A, C, D, and E (n = 2423). These variables included gender, region lived, household description, (χ^2^ ≥ 12.872, DF 6–33, *p* ≤ 0.05) owner age, education level, and household income in the last year (KW ≥ 12.27, *p* ≤ 0.001).

Therefore, despite the stratified sampling method used, there were meaningful and statistically significant differences between subpopulations surveyed at the same time point. Due to these differences and the missing data, it was considered inappropriate to combine the responses from the subpopulations to answer some research questions. Therefore, for in-depth analysis, only responses from the original cohort (subpopulation A) were examined. This subpopulation contained the most responses at all three time points and allowed for repeated measures analysis. However, to provide additional context for relevant research questions including examples of how the market research agency may have reported their findings, entire subpopulation results are reported as appropriate. All time points and subpopulations selected for reporting and analysis are clearly stated.

### 2.8. Statistical Analysis

Most data collected were at nominal or ordinal levels, and the data that were interval or ratio level were not normally distributed; therefore, non-parametric tests were employed throughout. Descriptive statistics were presented using frequencies and percentages, medians and interquartile ranges, or means and standard deviations as appropriate. For repeat measures, change scores were calculated. For any ordinal data that included “don’t know” or “not sure” responses, these were removed before analysis. Between-group differences were tested using Chi-Squared (χ^2^), Mann–Whitney U (U), Kruskal–Wallis (KW), and Wilcoxon Signed Rank tests. Correlations were tested using Spearman’s rho, and binomial repeat measures changes were tested using McNemar’s (exact) test.

### 2.9. Extracted Variables

Key variables extracted from survey responses are presented in Table 2. These variables were derived to achieve the aims and answer each research question.

### 2.10. Research Questions

#### 2.10.1. Aim 1: Establishing How Effective the Campaign Was at Reaching the Target Audience

To establish how effective the campaign was at reaching an audience, gender, age, region of the UK where respondents lived, highest education level, and household income in the last year were examined. Differences in demographics between those who had and had not heard of the campaign were explored. For those that had reported hearing about the campaign, differences in the way they had heard about and engaged with it were investigated. The target age group was also compared to the other age groups and the following questions were examined: Six months after the launch of #DogKind, what proportion of the sampled population had heard about the campaign? Of those that had heard about the campaign, how did they find out about it? Was there a difference in the demographics of the sampled population who had heard of #DogKind (and those that had not)? Of those that had heard about the campaign, how did they engage with it?

#### 2.10.2. Aim 2: Exploring How Effective the Campaign Was at Raising Awareness of SRB in Dog Owners

To explore how effective the campaign was at raising awareness of SRB in dog owners, within-subjects changes were calculated following exposure to the campaign and the following research questions were examined: Was there a change in respondents’ awareness of SRB? Was there a change in how long respondents thought the average dog could routinely be left alone, and as an occasional maximum, and was there a change in how long they reported leaving their dog alone, and as an occasional maximum? Similarly, was there a change in how happy respondents thought their dog was being left alone and how much they thought SRB impacts a dog’s happiness and wellbeing following the campaign?

#### 2.10.3. Aim 3: Exploring How Effective the Campaign Was at Increasing the Number of Dog Owners Intending to Seek Help for SRB from Reliable Sources

To explore how effective the campaign was at increasing the number of dog owners intending to seek help for SRB from reliable sources, within-subjects changes were calculated, and the following research questions were examined. Was there a change in the rating of how important it was to seek help for a dog suffering from SRB and in how likely respondents were to report intending to seek help if their dog showed signs of SRB following exposure to the campaign? Also, was there a change in respondents seeking help for SRB from different sources following the campaign?

#### 2.10.4. Aim 4: Determining Whether a Video Resource as Part of a Web-Based Campaign Increased the Effectiveness of the Campaign Objectives

To determine whether a video resource as part of a web-based campaign increased the effectiveness of the campaign objectives, differences between those who had and those who had not seen the video were tested. The following research questions were examined: Was there a difference in how long respondents thought the average dog could be routinely left alone and as an occasional maximum; how happy respondents thought their dog was being left alone; how much respondents thought SRB impacted a dog’s happiness and wellbeing; ratings of how important it was to seek help for a dog suffering from SRB; and the percentage of respondents intending to seek help from different sources between those who watched a video resource and those who did not?

## 3. Results

A total of 2002 respondents completed the survey at time point 1 or just before the launch of the campaign (subpopulations A and B) and 2422 (subpopulations A, C, D, and E) at time point 2 or six months after the launch of the campaign. Those in subpopulation A (n = 988) were more likely to be male (50.2%), live in London (18.4%), be aged 45–54 (median = 4 (25th percentile = 2, 75th percentile = 4), have an education level of A-levels or equivalent (4, (2, 4)), and have a household income of GBP 30–39,999 per year (4 (3,7)). The demographics of the different subpopulations are shown in Table 3.

### 3.1. Aim 1: Establishing How Effective the Campaign Was at Reaching the Target Audience

Six months after the launch of #DogKind, of the 1993 respondents who completed the survey, 29.4% had heard of the campaign. However, when these responses were explored by subpopulation, only 21.0% of subpopulation A (n = 988) and 26.7% of subpopulation C (n = 584) reported having heard of the campaign, whereas 52.7% of subpopulation E had heard of #DogKind (n = 421) (subpopulation B were non-respondents and D were missing data). The differences were statistically significant (χ^2^(2) = 146.56, *p* < 0.001).

Of the 207 respondents (21%) from subpopulation A (n = 988) who had heard about the campaign, most had heard via Facebook (47.3%) or the RSPCA website (41.7%). Similar trends were seen in subpopulations C and E. There were statistically significant differences between the proportion of the three subpopulations that reported hearing about the campaign via six of the ten media formats, these were the RSPCA newsletter, Twitter, Instagram, Google or another search engine, RSPCA stall at an event, and an RSPCA clinic or rehoming centre (Table 4). This provides further evidence of differences between the subpopulations.

When the 207 respondents in subpopulation A (n = 988) who had heard of #DogKind were compared to the 781 who had not, statistically significant differences were found in all five demographic variables tested. Those who reported having heard of the campaign were more likely to be male (65.2% vs. 46.2%) and living in specific geographical areas (e.g., more lived in London, 44.0% vs. 11.7%) compared to those that had not. They tended to have a higher education level than those who had not (U = 52 062.5, *p* < 0.001, 4 (3, 5) vs. 3 (2, 4)), as well as a higher household income (U = 11 0413, *p* < 0.001, 7 (5, 10) vs. 4 (3, 6)), and also tended to be younger (U = 43 898, *p* < 0.001, 3 (2, 4) vs. 4 (3, 5)). When comparing the campaign’s target age group of 25–34-year-olds to all the other age groups, those in the target age group were more likely to have heard about the campaign (48.9%) compared to those in the other age groups (16.5%, χ^2^(1) = 73.69, *p* < 0.001).

Of the 207 respondents in subpopulation A who had heard of #DogKind, 81.6% reported that they had engaged with the campaign, and they reported engaging in a mean of 2.05 ways (standard deviation (±) 1.61). Respondents most commonly had watched a video on the RSPCA website (53.8%), had seen or shared a Tweet (40.2%), or read the information on the RSPCA website (35.5%) (Table 5). The 169 (17.1%) respondents in subpopulation A who had heard of, and engaged with, the campaign reported engaging in a mean of 2.51 (±1.42) ways. There was no significant difference between those in the target age group compared with those who were in the other age groups in the number of engagement methods (U = 3499.5, *p* = 0.49).

### 3.2. Aim 2: Exploring How Effective the Campaign Was at Raising Awareness of SRB in Dog Owners

Before the campaign started in March 2019, 52.5% of all respondents who completed the baseline survey (subpopulations A and B, n = 2002) reported they had heard of SRB; six months after the launch of the campaign, this was 56.8% (subpopulations A, C, and D, n = 2001). When comparing the 998 subjects (subpopulation A), before the campaign 52.4% had heard of SRB, and six months after the launch of the campaign this had increased to 58.1% (n = 988, McNemar = 13.27, *p* < 0.001). The change was a result of 142 respondents who had not heard of SRB before the campaign reporting hearing of it following the campaign. However, 86 respondents who reported having heard of SRB before the campaign reported not having heard post-campaign. Interestingly, of those who had heard of the campaign, there was no significant difference in the proportion who reported having heard of SRB before and after the campaign (n = 207, McNemar = 0.37, *p* = 0.54). However, of those who had not heard of the campaign, there was a significant difference (n = 781, McNemar = 19.46, *p* < 0.001); before the campaign, 45.5% had heard of SRB, and six months after the launch this had increased to 53.3%. This change was a result of 123 respondents who reported not having heard of SRB before the campaign reporting they had, following the campaign, and 62 respondents who had heard of SRB before the campaign reporting not having heard post-campaign. This suggests the increase in awareness of SRB for subpopulation A was more likely to be because of a change in those who had not reported hearing of the campaign rather than those that had. There was also no difference between the target age group of 25–34-year-olds and the other age groups in a change of awareness of SRB (n = 66, McNemar (1) = 0.13, *p* = 0.73).

Before the campaign, respondents in subpopulation A thought that the average dog could be routinely left alone for a mean time of 5.88 h (±4.14) and as an occasional maximum for a mean time of 11.77 h (±12.05). Six months after the launch of the campaign, those same respondents reported a routinely left mean time of 6.08 h (±4.36) and an occasional maximum mean time of 12.38 h (±12.88). Neither of these were a significant difference (n = 988, mean change score = 0.18 ± 4.32; z = 151,748.5, *p* = 0.35; mean change score = 0.19 ± 12.61, z = 166,034, *p* = 0.13). There was also no significant difference between those who had seen the campaign and those who had not (n = 988, U = 85,311, *p* = 0.22; U = 80,593, *p* = 0.95). Neither were there any significant differences between the responses of the target age group of 25–34-year-olds and all other age groups (n = 207, U = 4428.5, *p* = 0.57; U = 4654.5, *p* = 0.99).

Similarly, respondents in subpopulation A reported leaving their dog alone on an average day for a mean time of 5.25 h (±5.62), and as an occasional maximum for a mean time of 7.44 h (±8.07). Six months after the launch of the campaign, those same respondents reported a routinely left mean time of 5.17 h (±5.57) and an occasional maximum mean time of 7.96 h (±8.59). Routinely left time alone was not significantly different (n = 988, z = 121,560, *p* = 0.33) but occasional maximum time was (n = 988, mean change score = 1.03 ± 8.26, z = 156,451.5, *p* = 0.03). From this subgroup, 232 respondents reported no change, whilst 343 reported reducing the number of hours their dog was left alone as an occasional maximum, whereas 413 respondents reported increasing the number of hours their dog was left alone as an occasional maximum. There were no significant differences between those who had seen the campaign and those who had not (n = 988, mean change score = 0.10 ± 5.28, U = 81,826, *p* = 0.78; U = 87,808, *p* = 0.05), nor between the responses of the target age group of 25–34-year-olds and the other age groups (n = 207, U = 4341.5, *p* = 0.44; U = 4328.5, *p* = 0.42).

Before the launch of the campaign, survey respondents in subpopulation A rated their dog as having a mean score of 6.08 (±2.41) on the “happiness scale” when being left alone; six months after the launch of the campaign, those same respondents reported a mean score of 6.22 (±2.37). This was a significant difference (n = 988, mean change score = 0.04 ± 2.11, z = 132,052.5, *p* = 0.05). From this subgroup, 291 respondents reported no change, 324 reported their dog feeling less happy when left alone, whilst 373 respondents reported their dog being happier when left alone after the campaign. There were no significant differences in happiness scale change scores, between those who had seen the campaign and those who had not (n = 988, U = 77,937.5, *p* = 0.42), nor between the target age group of 25–34-year-olds and the other age groups (n = 207, U = 5190.5, *p* = 0.17).

Equally, respondents in subpopulation A thought that SRB impacted a dog’s happiness and well-being with a mean score of 3.15 (±1.11) before the campaign, six months after the launch of the campaign, those same respondents reported a mean score of 3.21 (±1.08). This was a significant difference (n = 988, mean change score = 0.06 ± 0.78, z = 33,622, *p* = 0.03). From this population, 643 respondents reported no change, 156 reported SRB impacting a dog’s happiness less and 189 reported it impacting more. There were no significant differences in change scores between those who had seen the campaign and those who had not (n = 988, U = 79,435, *p* = 0.65), nor between the responses of the target age group of 25–34-year-olds and the other age groups (n = 207, U = 3644, *p* = 0.004, 3 (3, 4) vs. 3 (3, 4)).

### 3.3. Aim 3: Exploring How Effective the Campaign Was at Increasing the Number of Dog Owners Intending to Seek Help for SRB from Reliable Sources

Before the launch of the campaign respondents in subpopulation A rated the importance of seeking help for a dog suffering from SRB on average at 4.42 (±0.72) (n = 924), six months after the launch of the campaign, those same respondents reported the same mean score of 4.42 (±0.76) (n = 935, z = 21,463, *p* = 0.46). There were no significant differences in rating scale change scores (mean change score = 0.04 (±0.80)), between those who had seen the campaign and those who had not (n = 988, U = 82,787, *p* = 0.53), nor between responses of the target age group of 25–34-year-olds and the other age groups (n = 207, U = 4266, *p* = 0.26). However, respondents reported how likely they would be to seek help if their dog showed signs of SRB on average 3.15 (±0.82) (n = 899); six months after the launch of the campaign, those same respondents reported a significantly higher score (z = 28,110, *p* = 0.04, mean change score 3.19 (±0.78)). From this population, 531 respondents reported no change, 174 reported being more likely to be intending to seek help if their dog showed signs of SRB, and 142 reported being less likely. There were no significant differences, between those who had seen the campaign and those who had not (n = 988, U = 85,605, *p* = 0.15). However, there were significant differences when comparing the responses of the target age group of 25–34-year-olds to all other age groups (n = 207, U = 3968, *p* = 0.05, 3 (3, 4) vs. 3 (3, 4)).

Of all the respondents in subpopulation A (n = 988), the greatest number reported seeking help for SRB from a vet both before and after the campaign (29.5% and 31.4%). There was no significant difference in respondents’ reports of having sought help before and after the campaign from any other sources (n = 988, McNemar ≤ 3.05, *p* ≥ 0.08) apart from books. Before the campaign, 12.5% of respondents had sought help from books; six months after the campaign, this had increased to 14.9%. This change was a result of 94 more respondents reporting they had sought help from books after the campaign. However, the number of respondents who reported they had not sought help from books also increased by 67 after the campaign (McNemar = 4.20, *p* = 0.04).

The greatest number of respondents in subpopulation A reported that they would seek help for SRB from a vet both before and after the campaign (63.2% vs. 61.9%). There was no significant difference between the two time points (n = 988, McNemar ≤ 2.67, *p* ≥ 0.10) except for behaviourists. Before the campaign, 41.4% of respondents reported that they would seek help from a behaviourist; six months after the campaign, this had increased to 45%. This change was a result of 177 respondents reporting they would seek help from a behaviourist after the campaign but not before. However, 141 respondents reported they would before but not afterwards (McNemar = 3.85, *p* = 0.05).

When considering just the respondents in subpopulation A who had heard of the campaign (n = 207), the greatest number had reported seeking help for SRB from a vet both before and after the campaign (50.7% vs. 56.5%). There were no significant differences between where respondents reported having sought help from all sources before and after the campaign (n = 207, McNemar ≤ 2.35, *p* ≥ 0.13). The majority reported that they would seek help for SRB from an accredited behaviourist (50.7%) or a trainer (50.7%). This was shortly followed by a vet (49.3%), an accredited trainer (48.3%), a behaviourist (47.3%), and a vet nurse (47.3%). There was no significant difference between the proportion of respondents who reported that they would seek help before and after the campaign from any of the sources (n = 207, McNemar ≤ 2.75, *p* ≥ 0.10) except for vet nurses and Instagram. Before the campaign, 47.3% of respondents reported that they would seek help from a vet nurse; six months after the campaign this had increased to 58.5%. This change was a result of 59 more respondents reporting they would seek help from a vet nurse after the campaign, but also included 39 respondents who reported they would no longer seek help from a vet nurse after the campaign (McNemar = 5.10, *p* = 0.02). Equally, before the campaign, 33.8% of respondents reported that they would seek help from Instagram; six months after the campaign this had increased to 44%. This change was a result of 59 more respondents reporting they would seek help from Instagram after the campaign, but also included 38 respondents who reported they would no longer seek help from Instagram after the campaign (McNemar = 4.12, *p* = 0.04).

When comparing the responses of those who had heard of the campaign to those who had not, there were significant differences in the proportion that reported having or would seek help for each of the key sources listed (Table 6). Those who had heard of the campaign were significantly more likely to have sought help from a vet and reported they would seek help from a vet nurse. When comparing the responses of those in the target age group to the other age groups, there was one significant difference in where they reported having sought help. Overall, 68.2% of the target age group reported having sought help from a vet compared to 51.1% in the other age groups (χ^2^(1) = 5.36, *p* = 0.02). Differences between the target age group and other age groups for other sources of help were not significant (χ^2^(1) < 2.87, *p* > 0.09).

### 3.4. Aim 4: Determining Whether a Video Resource as Part of a Web-Based Campaign Increased the Effectiveness of the Campaign Objectives

To establish whether a video resource as part of a web-based campaign increased the effectiveness of the campaign objectives, the results from time point 3 (eight months after the launch of the campaign) and subpopulation A (n = 190) were explored. These respondents had recently viewed the web pages and been pseudo-randomly allocated to either the video resource or the campaign web pages for the study. Of this population, 39.5% reported seeing the video, 47.9% reported not seeing the video, and 12.6% were not sure. To ensure the effectiveness of the video was being measured, those who reported being “not sure” whether they had seen a video or not were removed from further analysis. This resulted in a sample size of 75 reporting seeing the video and 91 not.

There were no significant differences between those who watched a video and those who did not; how long respondents thought the average dog could be left alone routinely (U = 3421.5, *p* = 0.98); or as an occasional maximum (U = 3037.5, *p* = 0.22); how happy respondents thought their dog was being left alone (U = 3751.5, *p* = 0.27); how much respondents thought SRB impacted a dog’s happiness and wellbeing (U = 6042, *p* = 0.06); how important it was to seek help for a dog suffering from SRB (U = 6623.5, *p* = 0.89); and where respondents reported they would seek help following the campaign (χ^2^(1) < 0.98, *p* > 0.32).

## 4. Discussion

A total of 2002 respondents completed the pre-survey (subpopulations A and B) and 2422 (subpopulations A, C, D, and E) completed the six-month survey. Differences between the subpopulations were found, therefore only subpopulation A was used for further analysis. The campaign reached nearly 30% of the sampled population and was effective in targeting the intended age group of 24–34 years old. However, there was very limited change in raising awareness of SRB and seeking help from reliable sources following exposure to the campaign. The addition of the video showing dogs demonstrating signs of SRB did not make any measurable difference to the campaign objectives.

Before exploring the findings in more depth, the sampling approach must be considered to put subsequent discussions into context. Despite the market research agency stating that the sampled populations were equivalent, it was clear that there were significant and meaningful differences between them. The approach taken to sampling resulted in less than half the specified total of 2000 respondents providing data that could be directly compared pre- and post-campaign. This problem with the sampling and subsequent data set may have resulted in the findings presented by the market research agency to the RSPCA giving an inaccurate portrayal of the impact of the campaign. It should also be noted that this comes with an economic impact on the charity in so much as there was payment for data that could not be fully utilised. Therefore, it is essential to highlight the impact that the sampling approach and missing data sets have had on the evaluation and could potentially have on charities evaluating future campaigns, who may take the data provided by market research agencies at face value.

It is also necessary to question whether the approach taken by any market research company using panels is representative of the specified population in the UK that they seek to understand. The market research agency sampled their “panellists” who are individuals who complete surveys regularly for “incentive points” and who may have a different approach to completing surveys than the general population who do not subscribe to these activities. It could be queried whether these more experienced and incentivised survey completers consider their answers and/or reply honestly to the questions provided. Their motivations for completing surveys may well impact their responses and the overall results. This can be demonstrated in changes in response from those at pre-survey and the six-month survey. By asking people about their understanding of SRB before the launch of the campaign, this should have meant that everyone reported having heard of it six months later, which was not the case. Similarly, many reported that they had heard of SRB before the campaign but reported not hearing about this six months later. This could mean that they had either forgotten or were simply responding more randomly and without due consideration for the answers given. If this is the case, the reliability of all answers should be questioned.

Whilst using panellists can avoid the problems of self-selected survey samples including a disproportionate number of keen owners, they evidently present potential issues. Therefore, for future campaign evaluations, we recommend that if data are being collected by a third party, information regarding sampling strategies and anticipated response rates are provided, and care is taken to avoid survey fatigue and to spot check the validity of responses by having some respondents repeat their entry and the consistency of their responses compared.

### 4.1. How Effective Was the Campaign at Reaching the Target Audience?

The overall results showed that 29.4% of respondents had heard about the campaign six months after its launch. Whilst there are few equivalent published companion animal welfare campaigns with which to compare our findings, the campaign reached nearly a third of its intended population. As the most recognised animal welfare charity in the UK, the RSPCA brand may have contributed to the extent to which the campaign was recognised [4]. The most effective media for people hearing about the campaign were Facebook and the RSCPA website. This is to be expected given that the use of digital media in both society and campaigns is now the norm.

Differences in the demographics of those who had heard about the campaign compared to those who had not could be explained by the previously discussed sampling approaches and population differences, but also by the profiles of people who typically follow and support the RSPCA. Those who had heard of the campaign were more likely to be male, living in London, have a university degree or equivalent, and have a household income of GBP 60–69,999. Interestingly, however, those in the target age group of 25–34-year-olds were more likely to have heard about the campaign than those in other age groups. As the campaign was created following the findings of the #DogKind Report [42] and was further developed through a series of focus groups commissioned by the RSPCA to target this age group, this aspect of the campaign was successful.

Whilst the most effective media for people hearing about the campaign were Facebook and the RSCPA website, this was not consistently the case. Those living in London were most likely to have heard via the website or Instagram whilst those in the East and West Midlands were more likely to have heard via posters or leaflets; all other means by which people had heard about the campaign were not significantly different across the regions. Those differences in how people had heard of the campaign may be due to variations in the marketing approach taken by the RSPCA or simply to regional differences. Regional differences have been found in health-related nationwide campaign evaluations such as those promoting reduced salt intake [46] and for activities such as smoking cessation; regional differences can also be linked to socioeconomic status [7]. These findings demonstrate that a targeted approach is required when planning campaigns aimed at specific demographic groups, and that there may be value in improving materials disseminated by the less popular media to reach a wider demographic whilst maintaining the media (e.g., Facebook) demonstrated to be currently most effective.

### 4.2. How Effective Was the Campaign at Raising Awareness of SRB in Dog Owners?

The charity hoped that the campaign would increase awareness of SRB and overall figures showed there was a significant increase in the proportion of respondents reporting hearing of SRB six months after the campaign was launched. However, when the results were explored in further detail, they showed the increase in awareness of SRB was because of more of a change in those who had not reported hearing of the campaign rather than those who had. Equally, the use of terminology and questions about SRB in the baseline survey is likely to have led to an increase in awareness of SRB reported in subsequent surveys, so some increased awareness may have been due to the survey rather than the campaign. Changes in awareness of SRB in those who had not heard of the campaign (and those who had) may have been because of uncontrolled confounding variables such as individuals’ personal experiences, exposure to other campaigns or media discussing SRB, or social contagion over the six months between the pre-survey and six-month survey. Equally, there was no significant difference between the target age group when compared to all other age groups, so although the RSPCA were successful in ensuring this age group heard about the campaign, delivering the key message was less successful.

It was anticipated that the campaign should increase awareness of some of the problems with leaving dogs alone and reduce the time respondents thought that the “average” dog could be left and potentially trigger a change in behaviour of how long respondents left their dogs alone. However, of the “time alone” variables tested, there was only one significant change, and this was in how long respondents reported leaving their dog for an occasional maximum. Following the campaign, respondents reported increasing the time their dog was left as an occasional maximum from a mean time of 7.44 h (±8.07) to 7.96 h (±8.59) after the campaign. This increase clearly goes against the campaign message. There were no changes in how long respondents left their dog alone on a routine basis nor in how long they thought the “average” dog could be left routinely or as an occasional maximum.

Respondents thought that the average dog could be left for nearly six hours per day and reported leaving their dogs for an average of 5.25 h per day. These figures are concerning given that most dog welfare charities, including the RSCPA, recommend a dog is left for no more than a maximum of four hours per day or less depending on the individual dog’s needs [42]. Equally concerning was the fact that respondents thought the average dog could be left as an occasional maximum for around 12 h and many reported leaving their dog for nearly eight hours on occasion, going very much against current guidance.

As the campaign highlighted the signs of SRB, it was anticipated that respondents may notice these signs of “unhappiness” in their dogs, following exposure to the campaign and therefore revealing problems that they were previously unaware of. Interestingly, there was a significant increase from a mean score of 6.08 (±2.41) to 6.22 (±2.37) in respondents’ rating of their dog’s happiness when being left following the campaign. This suggests either that the campaign did not increase respondents’ awareness of SRB signs in their dogs, or that they had changed something to make their dogs happier when they left them. It is possible that despite not significantly reducing the time they left their dogs, respondents may have taken on board some of the advice presented in the campaign on “treating” SRB or “teaching their dog how to cope on their own” [47]. There was a difference in how much respondents thought SRB impacts a dog’s happiness more generally, with a significant increase detected. However, there was no significant difference between those who saw the campaign and those who did not, making it difficult to attribute this change to the campaign.

### 4.3. How Effective Was the Campaign at Increasing the Number of Dog Owners Intending to Seek Help for SRB from Reliable Sources?

One of the aims of the campaign was to increase the number of dog owners intending to seek help for SRB and to encourage dog owners to seek help from a vet initially but then, if appropriate, consult a registered behaviourist. Post-campaign, there was no change in respondents’ rating of the importance of seeking help for a dog showing signs of SRB, though there was a significant increase in the proportion of respondents intending to seek help for their dog if they showed signs of SRB (mean change score 3.19 (±0.78)). There were no significant differences between those who reported having seen the campaign and those who had not, making it difficult to attribute the change to the campaign.

Most respondents reported that they would or had sought help from a vet if their dog displayed signs of SRB, which supports the primary campaign message. However, there were no significant changes in where people sought help after the campaign except for books where there was an increase. This finding may be a spurious result due to multiple testing but may also show the campaign did not do as predicted in terms of seeking help from an accredited behaviourist. However, when focusing on just those who had heard of the campaign, the majority reported that they would seek help from an accredited behaviourist or trainer, shortly followed by a vet, accredited trainer, behaviourist, and vet nurse. But these findings did not change significantly after the campaign except for an increase in the proportion intending to seek help from vet nurses, which is contradictory to the campaign message and once again suggests that the campaign did not do as predicted. The potential impact of social desirability bias in responses such as these must also be considered; respondents may wish to be seen to do the “right thing” and not necessarily answer truthfully. What is more, it is noteworthy that successful human health campaigns often chart success over the years, so it may not be realistic to expect statistically significant changes within a single campaign. The campaign may have helped to raise awareness and change attitudes beyond the capacity of this evaluation to capture. More research is likely required to better understand this in animal welfare campaigns. However, based on the results of this study alone, we would suggest that the methods used in this campaign have limited effectiveness and hence other approaches to changing public attitude and behaviour should be considered.

### 4.4. Did a Video Resource as Part of a Web-Based Campaign Increase the Effectiveness of the Campaign Objectives?

Due to the nature of SRBs, they are not commonly observed directly by the pet’s owner. Until the owner sees the level of distress shown by their animal directly, they may dismiss the significance of this behavioural disorder. It was therefore anticipated that a video showing dogs displaying signs of SRB may increase the effectiveness of the campaign objectives. However, the use of a video as part of the campaign did not make any difference to any of the outcome variables tested, so we can conclude that, in the short term, the video was not effective at increasing the messaging of the campaign. This may be a result of the specific video content or the effectiveness of video more generally [22]. These findings also contradict the findings of a recent scoping review looking at the effectiveness of video in mental health promotion [23]. Limitations of this element of the study were not only the small sample size, but the lack of a way to check whether respondents had viewed the video or read the information on the web pages. We relied on respondents reporting whether they had watched a video or seen the webpages and with 12.6% of respondents reporting that they were not sure, the attention paid by respondents to the resources may have an impact on these findings. However, our findings do suggest that in its current form, the video used is not an effective education tool. An alternative format of information delivery or more interactive online content may be better at changing behaviour, and hence we suggest that future campaigns should consider alternative approaches as part of their education delivery strategy.

The complex interaction between knowledge and understanding, intention, and behaviours should also be considered when discussing this study’s findings. Knowledge and/or intention do not necessarily result in the desired behaviour. This contradiction in dog owners has been highlighted in studies by Rohlf et al. [48] and Westgarth et al. [49], who have shown that even owners who identify as being responsible do not always demonstrate this in their behaviours. This may offer some explanation as to why the campaign may not have changed people’s intended behaviours. Additionally, the intricacy of each individual’s situation will not have been captured through this study, and extraneous variables such as changes in work patterns, socioeconomic status, health, etc. may have impacted the study outcomes. Due to these factors and the study’s limitations, as described above, no causal effects (or lack of) can directly be attributed to this campaign.

## 5. Conclusions

Overall, the campaign appeared to be effective in reaching its target audience of 25–34-year-olds but not at raising awareness of SRB or increasing the number of dog owners intending to seek help from reliable sources. The inclusion of the video in the campaign made no difference in its effectiveness. This study shows that this campaign had limited success in achieving its targets, despite the market research agency’s overall or “headline” results making the campaign appear “successful”. This rigorous evaluation has shown that those initial findings do not represent the whole picture nor give the level of detail required to understand what aspects of the campaign worked and what did not. We recommend that whilst the idea of targeting a specific demographic and choosing media accordingly is valuable, the development of more interactive resources and testing the effectiveness of potential education materials, prior to investing in wider dissemination, may be a more cost-effective approach.

## Figures and Tables

**Table 1 animals-14-00484-t001:** Sample sizes and data available for the different subpopulations sampled.

Subpopulation	n	Number Completing the Pre-Survey or before the Launch of #DogKind	Number Completing the Survey Six Months after the Launch of #DogKind	Number Completing the Survey Eight Months after the Launch of #DogKind; Webpages Only	Additional Information
A	988	988	988	190	Six-month survey first cohort respondents
B	1014	1014	0	0	Six-month survey first cohort non-respondents
C	584	0	584	79	Six-month survey second cohort respondents with full data
D	429	0	429	0	Six-month survey second cohort respondents with missing #DogKind responses due to lack of consent to share personally identifiable data with third parties
E	421	0	421	0	Six-month survey third cohort with missing dog demographics, ownership practices, and SRB knowledge responses due to system error
Total	3437	2002	2423	269	

**Table 2 animals-14-00484-t002:** Variables tested, including timepoint, survey question, answer options, and variable description with coding.

Time Points	Survey Category	Survey Question	Survey Answer Options	Variable Name	Coding	Level of Measurement
1 and 2	Respondent demographics	What is your gender?	Male	Gender	1	Nominal
			Female		2	
			Non-binary		3	
		Which of the following categories best describes your age?	18–24	Age	1	Ordinal
			25–34		2	
			35–44		3	
			45–54		4	
			55–64		5	
			56–74		6	
			75–84		7	
			85 and over		8	
		Which region to you live in?	East Midlands	Region	1	Nominal
			East/East Anglia		2	
			London		3	
			North East		4	
			North West		5	
			Northern Ireland		6	
			Scotland		7	
			South East		8	
			South West		9	
			Wales		10	
			West Midlands		11	
			Yorkshire and the Humber		12	
		What is your highest education level?	No formal qualifications	Highest education level	1	Ordinal
			GCSEs or equivalent		2	
			A levels or equivalent		3	
			University degree or equivalent		4	
			Postgraduate degree or equivalent		5	
			Prefer not to answer			
		Approximately what was your household income in the last year?	Less than GBP 10,000		1	Ordinal
			GBP 10,000 to GBP 19,999		2	
			GBP 20,000 to GBP 29,999		3	
			GBP 30,000 to GBP 39,999		4	
			GBP 40,000 to GBP 49,999		5	
			GBP 50,000 to GBP 59,999		6	
			GBP 60,000 to GBP 69,999		7	
			GBP 70,000 to GBP 79,999		8	
			GBP 80,000 to GBP 89,999		9	
			GBP 90,000 to GBP 99,999		10	
			GBP 100,000 to GBP 199,999		11	
			GBP 200,000 to GBP 299,999		12	
			More than GBP 300,000		13	
			Prefer not to say			
1 and 2	Current ownership practices	On an average day in the last month, how many hours did your dog spend at home without human company?	Time in hours	Time alone		Ratio
		What was the longest single period your dog was left alone without human company in the last month?	Time in hours	Occasional maximum		Ratio
1 and 2	Dog behaviours and seeking help	On a scale of 1–10, how happy do you think your dog is being left alone without human company?	1 (unhappy)−10 (happy)	Happiness scale		Interval
		Where have you or would you go for help if needed? Vet, vet nurse, behaviourist, accredited behaviourist, trainer, accredited trainer, friends or family, Google or another search engine, Facebook, Twitter, Instagram, YouTube, blog, books, and other	Have sought help from Y/N Would seek help from Y/N	Source of help		Nominal
		If you knew your dog was barking, whining, or howling when you were out, how likely would you be to seek help?	Very likely	Likely to seek help for dog showing signs SRB	4	Ordinal
			Likely		3	
			Unlikely		2	
			Very unlikely		1	
			I don’t know		0	
		How important do you think it is to seek help for a dog suffering from separation anxiety?	Very important	Importance of seeking help	5	Ordinal
			Somewhat important		4	
			Neither important nor unimportant		3	
			Somewhat unimportant		2	
			Not very important		1	
			I don’t know		0	
1 and 2	Knowledge and understanding of SRB	How long do you think the average dog can routinely be left alone without human company?	Time in hours	Average dog routinely left		Ratio
		How long do you think as an occasional maximum the average dog can cope with being left at home without human company?	Time in hours	Average dog occasional maximum		Ratio
		Have you heard of SRB or separation anxiety?	Yes	Heard of SRB		Nominal
			No			
			Not sure			
		How much do you think separation anxiety impacts on a dog’s happiness and wellbeing?	Very much	Impact SRB on happiness and wellbeing	4	Ordinal
			Somewhat		3	
			Not much		2	
			Not at all		1	
			I don’t know		0	
2 and 3	#DogKind campaign	Have you heard of the RSPCA’s #DogKind campaign?	Yes	Heard of campaign		Nominal
			No			
		How did you find out about the RSCPCA’s #DogKind campaign?	RSPCA website	Media format heard of campaign		Nominal
			RSPCA newsletter			
			Facebook			
			Twitter			
			Instagram			
			Google or another search engine			
			RSPCA stall at an event			
			RSPCA clinic or rehoming centre			
			Posters or leaflets			
			Can’t remember			
		Have you actively engaged with the campaign?	Yes	Engaged with campaign		Nominal
			No			
		How did you engage?	Read information on the website	Engagement methods		Nominal
			Watched video on the website			
			Seen and shared tweet(s)			
			Seen and shared Facebook post(s)			
			Read blogs			
			Picked up a leaflet			
			Given a leaflet to someone I know			
			Visited a RSPCA stall at an event to discuss SRB			
			Visited a RSPCA clinic or rehoming centre and discussed SRB			
			Told someone I know about the campaign			
		How many ways did you engage with the campaign?	Number of methods reported to have engaged with campaign out of maximum of ten options	Number of ways engaged with campaign		Ratio
		Have you visited the webpages?	Yes	Visited webpages		Nominal
			No			
		When you visited the RSPCA’s #DogKind campaign, did you see a video?	Yes	Seen video		Nominal
			No			
			Not sure			
		Since finding out about #DogKind, do you think you would be more or less likely to seek help if your dog was showing signs of separation anxiety?	Much more likely	How likely to seek help	5	Ordinal
			More likely		4	
			Not changed		3	
			Less likely		2	
			Much less likely		1	
			I don’t know		0	

**Table 3 animals-14-00484-t003:** Participant demographics for the different subpopulations’ data collected at baseline, before the launch of #DogKind, and data collected six months after the launch of the campaign.

		Recruited at Time Point 1 before the Launch of #DogKind				Recruited at Time Point 2 Six Months after the Launch of #DogKind					
		Subpopulation A n = 988		Subpopulation B n = 1014		Subpopulation C n = 584		Subpopulation D n = 429		Subpopulation E n = 421	
		Frequency	%	Frequency	%	Frequency	%	Frequency	%	Frequency	%
Gender	Male	496	50.2	586	57.8	307	52.6	204	47.6	179	42.5
	Female	492	49.8	426	42.0	276	47.3	224	52.2	241	57.2
	Non-binary	0	0	2	0.2	1	0.2	1	0.2	1	0.2
Region	East Midlands	64	6.5	40	3.9	38	6.5	29	6.8	33	7.8
	East/East Anglia	59	6.0	55	5.4	24	4.1	30	7.0	27	6.4
	London	182	18.4	338	33.3	145	24.8	44	10.3	56	13.3
	North East	74	7.5	70	6.9	28	4.8	20	4.7	19	4.5
	North West	110	11.1	106	10.5	61	10.4	52	12.1	53	12.6
	Northern Ireland	15	1.5	29	2.9	9	1.5	11	2.6	13	3.1
	Scotland	81	8.2	62	6.1	37	6.3	37	8.6	25	5.9
	South East	121	12.2	94	9.3	71	12.2	57	13.3	54	12.8
	South West	89	9.0	64	6.3	47	8.0	47	11.0	40	9.5
	Wales	38	3.8	39	3.8	40	6.8	30	7.0	26	6.2
	West Midlands	77	7.8	55	5.4	48	8.2	38	8.9	31	7.4
	Yorkshire and the Humber	78	7.9	62	6.1	36	6.2	34	7.9	44	10.5
		Median	IQR	Median	IQR	Median	IQR	Median	IQR	Median	IQR
Age		4	2	3	3	3	3	4	2	3	2
Highest education level		3	2	4	2	4	2	3	2	3	2
Household income in last year		4	4	6	5	5	5	4	3	4	3

**Table 4 animals-14-00484-t004:** Comparison of the percentage of each subpopulation who reported hearing about the campaign via each media format six months after its launch.

	Subpopulation A n = 207	Subpopulation C n = 156	Subpopulation E n = 222	Chi Squared Test for Differences between Subpopulations	
	%	%	%	Chi ^1^	*p*
RSPCA website	41.1	41.7	47.7	2.3	0.313
RSPCA newsletter	35.2	37.2	23.9	9.7	0.008
Facebook	47.3	55.8	47.7	3.1	0.213
Twitter	29.5	25.6	12.6	19.3	<0.001
Instagram	27.5	31.4	7.7	39.3	<0.001
Google or another search engine	16.9	23.7	9.9	13.1	0.001
RSPCA stall at an event	10.6	19.2	7.2	13.2	0.001
RSPCA clinic or rehoming centre	9.2	10.3	3.6	7.5	0.023
Posters or leaflets	2.9	5.1	6.3	2.8	0.249
Can’t remember	1.9	4.5	5.4	3.6	0.166

^1^ DF 2.

**Table 5 animals-14-00484-t005:** Percentage of subpopulation A who reported engaging with the campaign by each of the ten methods, six months after its launch.

	Subpopulation A n = 169
Watched video on the website	53.8
Seen and shared Tweet(s)	40.2
Read information on the website	35.5
Read blogs	30.8
Seen and shared Facebook posts	29.6
Picked up a leaflet	23.7
Given a leaflet to someone they know	14.8
Visited an RSCPA stall at an event to discuss SRB	12.4
Visited an RSCPA clinic or rehoming centre at an event to discuss SRB	6.5
Told someone they know about the campaign	3.6
Other	0

**Table 6 animals-14-00484-t006:** Differences between those who had heard of the campaign and those who had not (subpopulation A) in where respondents have sought help from and would seek help from following the campaign.

Source of Help	Have Sought or Would Seek Help	Heard of #DogKind n = 207	Not Heard of #DogKind n = 781	Chi Squared Test for Differences between Those Who Had Heard of the Campaign and Those Who Had Not	
		%	%	Chi ^1^	*p*
Vet	Have sought help	56.6	24.7	76.90	<0.001
	Would seek help	46.9	65.9	25.27	<0.001
Vet nurse	Have sought help	39.1	11.5	87.14	<0.001
	Would seek help	58.5	62.2	0.98	0.321
Behaviourist	Have sought help	32.9	5.2	127.00	<0.001
	Would seek help	48.8	44.0	1.49	0.222
Accredited behaviourist	Have sought help	38.6	3.1	219.87	<0.001
	Would seek help	53.6	38.5	15.31	<0.001
Trainer	Have sought help	31.4	6.0	104.89	<0.001
	Would seek help	51.2	45.1	2.48	0.115
Accredited trainer	Have sought help	29.5	3.8	128.52	<0.001
	Would seek help	55.6	41.7	12.64	<0.001

^1^ DF 1.

## Data Availability

All data are owned by RSPCA and have been shared with permission.

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
