# Peer review of "Do Animal Welfare Education Campaigns Really Work? An Evaluation of the RSPCA’s #DogKind Campaign in Raising Awareness of Separation-Related Behaviours in UK Dog Owners"

_animals, 2024, doi:10.3390/ani14030484_

Round 1

Reviewer 1 Report

Comments and Suggestions for Authors

This paper is well written and reasonably concise. It details an important but under-researched aspect of canine welfare and strongly highlights the need for good evaluation. The introduction provides a solid overview to the topic. This is aided by sufficient referencing. The methods provides a detailed account of data collection. The results summarise key information and comparisons and is aided by being grouped into 4 detailed aims. However, I found that having 5 unique populations across 3 surveys made this section slightly confusing at times. Aspects of this could potentially be clearer. The discussion addresses limitations – especially those relating to the sampling approach. It’s very refreshing (in some ways!) to read about issues faced and attempts to overcome these. Research studies are seldom – if ever – perfect (especially, in my experience, when external market research companies are involved!) and acknowledging this helps provide context and allows the reader to better appreciate the reliability and potential impact of this work. The conclusion is a fair summary of the study. It is also refreshing (again, in some ways!) to see a study focusing on what didn’t make a difference, e.g. this campaign was not successful in some of its aim and it is important to document this in print. This in itself is really useful, especially to those working in animal welfare charities involved in campaigns and education (although understanding of the sampling methods enables users to questions aspects of the study and better understand how meaningful such findings may be). All in all a very interesting and thought-provoking read!

I have provided a small number of specific comments below. These are minor and many reflect my personal preferences (or interest!) hence are suggestions and not all may necessarily require any changes.

Line 77: Use of commas at the start of the sentence is a little clunky.

Lines 88-89: Suggest condensing list of authors (e.g. Stead et al.).

Line 93: UK has already been used as an abbreviation in line 46. I’m not sure that the full “United Kingdom” is actually needed but if the authors want to include this, it should be after its first use.

Line 102: Check reference (this should be a number – 44? – rather than the author’s name, but if this is the correct ref #, the placing in the ref list is likely incorrect, hence #s need to change).

Line 126 & 128: Check “a” and “b” in references.

Line 135 (and throughout introduction): Use of “s” after SRB. Check that use of SRB/SRBs is correct and refers correctly to behaviour / behaviours.

Line 168: Why was this age group chosen?

Lines 172-177: What were the key take away messages from this campaign, e.g. where to go for help if SRB suspected, whether dogs should be left along and for how long, etc? You mention in the next p/g about “seeking help for SRB from reliable sources”: were these explicitly described?

Lines 186-191: Were these webpages available to the general public too (including the written and/or video info)?

Line 194: What survey software was used/was this through the market research company’s software?

Line 201: Are these meant to be representative of the general public/dog owners? Was there any attempt to reach more/only respondents within the selected age group?  Are you able to comment more on this/the market research company’s recruitment? (I’ve since noted this is discussed in the discussion but I think it would be useful to highlight the intention when discussing sampling.)

Line 205: How was 2,000 respondents decided upon? Was this a sample size calculation? Given that for most comparisons, a sample size of 2,000 was not available from both (or indeed all 3) time points, how might this impact findings?

Lines 220-228: I really like that you’ve outlined the challenges of recruiting participants, even when incentives are offered, and the issues that unfortunately we’ve all faced in terms of (incorrect) consent and data collection… it’s really good that this is documented! Was ~50% response rate (line 215) for the second survey in line with expectations (yours/the market research company)? Why/why not?

Lines 236-237: Did you know which participants were directed to the writing or video or were you reliant on them reporting this back to you? (You mentioned “pseudo-randomly”: can you provide any more details on this? Were there roughly equal numbers directed to both?) Were there any checks in place to ensure that participants watched the video in full?

Line 246+: Were all survey questions mandatory/optional?

Lines 248-249: Was the pilot within the market research company’s survey software? How many owners were involved? Can you provide more details about the changes based on feedback?

Lines 289-295 (and more generally throughout the results): Although I appreciate the reasons for including all data, focusing on subpopulation a would make the results easier to follow. Would there be any way of making it clearer where results relate just to subpopn a (and/or make it more obvious when other subpopns are being referred to)? I’ve found myself having to go back and forth a lot to recap which subpopn might be included where. E.g. Section 3.3 (lines 472-489): it’s not clear what subpopn this section refers to (presumably a). Possibly using capitalisation for the subpopns (i.e. A, B, C, D, E) might help these stand out more and not get so lost in the text?

Line 309: I appreciate that key variables are included in this table but would it be possible to provide complete questionnaire questions sets, e.g. as an appendix or supplementary file? (I’d suggest moving this table, e.g. to a supplementary file,  as a non-paper format might suit it better – although it’s helpful to have all of this information, it is a very large table and it is sometimes difficult to understand exactly what responses match with what question/how it would have been asked.) What additional questions were asked (only) in the third survey?

Line 312+: Might it help to have a distinct research question or description for each aim? (To help distinguish it from the other aims.)

Lines 383-393 (and more generally throughout the results): I don’t think all of these %s are shown in tables. Will these breakdowns be available as supplementary materials? Specifically, Line 387: I don’t think Table 5 shows this? It would be useful to have more detail, e.g. lines 388-389, especially as I think this is referred to – but by highest annual household income band, etc. in the discussion (lines 612-614)?

Line 396-397: This was not related to the survey directing respondents to the video? I.e. they had happened to see this? (Sorry, I’m finding this a little confusing to follow in places!)

Line 404: Could this table be ordered in descending order, e.g. highest % reported first, etc.

Lines 549-550: Similar as an earlier point, would you have expected to see a roughly 50/50 split? (Was the video easier to forget/did respondents just not bother watching it if they were after the survey incentive and there were no checks in place?) Were there any ways of checking whether respondents had seen the video or was this based on their self-reporting? Is this sample size larger enough to draw any firm conclusions?

Lines 619-625: Was there any online targeting of particular demographics, e.g. through FB ads? For the more face-to-face type methods (events, clinic, posters), were these focused on any particular areas (geographically) or aimed more towards particular demographics?

Line 666: Is “highlighting” correct here? (Would “highlighted” fit better?)

Overall, a really interesting paper! I look forward to seeing this published and hope it will lead to better accountability of market research companies, increased awareness of the need for good evaluation of campaigns and education projects, and further research into this important area.

Reviewer 2 Report

Comments and Suggestions for Authors

Thank you for the invitation to review this manuscript. This is important work as you rarely see an evaluation of animal charity campaigns or education programmes and so this fills an important gap in the field. I have provided some suggested edits below. Overall, I felt more attention could be paid to the lessons learnt from the evaluation process (beyond the sampling method), and how the RSPCA has adapted/changed since conducing this research in 2019. I think this would be helpful for other groups who are also trying to raise awareness of animal welfare through campaigns or similar activities.

Introduction

1)      Is it worth briefly discussing evaluation methods used for campaigns? As the evaluation strategy may actually impact whether a positive impact has been identified? E.g. could just be the way impact is measured

2)      Reference 10 – could be ‘et al’ in the text?

3)      Why could it be that previous evaluations have only found a modest / moderate impact/success? Worth discussing key reasons briefly here (and does the RSPCS campaign address the limitations in previous campaigns?) .

4)      What is the theory behind the effectiveness of video material? E.g. could draw from learning theories. This could build a stronger rationale for the inclusion in campaigns.

5)      Could add further rationale as to why SRBs are important to prevent (not just extreme cases of relinquishment).

Method

1)      Is it possible to make it a little clearer on how the sample was divided into test and control groups? I presume the control group has not engaged with the campaign and video, whereas those who has were the test group? Please make be really clear here. Also include the final N for test/control groups so easy for reader to understand.

2)      There is limited information regarding the survey measures. For example, it is said that attitudes were measured, but it doesn’t say how – e.g. was this a new measure, how was it developed? Was it based on an existing standardised attitude measure? How many items did the measure include? How was it coded etc.? How do you know it is assessing attitudes? I couldn’t also see a research question relating to attitudes, and no results regarding attitudes were presented? Seems the evaluation was based on awareness only? And willingness to seek support?

Results

1)      Proof read to ensure the results are not too wordy and repetitive as they can be hard to follow at times

2)      Table 2 could be in an appendix. Could text in table 3 be made smaller so the content fits easier on the page?

Discussion

1)      The discussion provides a good summary of the results but it is mostly a repetition of the result section. Less attention is paid to why these findings may have been found and what the next steps/actions will be. Expanding arguments here to discuss how each key finding can inform future practice is important and should be considered in detail.

2)      Need to consider and mention that the data was collected quite a few years ago in 2019 – what has the RSPCA done since / learnt from this research? What are the next steps? And again how did/will this evaluation impact future campaign work? How could this research inform future campaign work? What are your recommendations?
